# Role of Viruses in the Pathogenesis of Multiple Sclerosis

**DOI:** 10.3390/v12060643

**Published:** 2020-06-13

**Authors:** Rachael E. Tarlinton, Ekaterina Martynova, Albert A. Rizvanov, Svetlana Khaiboullina, Subhash Verma

**Affiliations:** 1School of Veterinary Medicine and Science, University of Nottingham, Loughborough LE12 5RD, UK; rachael.tarlinton@nottingham.ac.uk; 2Insititute of Fundamental Medicine and Biology Kazan Federal University, 420008 Kazan, Russia; ignietferro.venivedivici@gmail.com (E.M.); rizvanov@gmail.com (A.A.R.); 3School of Medicine, University of Nevada, Reno, NV 89557, USA; sv.khaiboullina@gmail.com

**Keywords:** multiple sclerosis, human herpesvirus 6, varicella–zoster virus, cytomegalovirus, John Cunningham virus, human endogenous retroviruses, Epstein–Barr virus

## Abstract

Multiple sclerosis (MS) is an immune inflammatory disease, where the underlying etiological cause remains elusive. Multiple triggering factors have been suggested, including environmental, genetic and gender components. However, underlying infectious triggers to the disease are also suspected. There is an increasing abundance of evidence supporting a viral etiology to MS, including the efficacy of interferon therapy and over-detection of viral antibodies and nucleic acids when compared with healthy patients. Several viruses have been proposed as potential triggering agents, including Epstein–Barr virus, human herpesvirus 6, varicella–zoster virus, cytomegalovirus, John Cunningham virus and human endogenous retroviruses. These viruses are all near ubiquitous and have a high prevalence in adult populations (or in the case of the retroviruses are actually part of the genome). They can establish lifelong infections with periods of reactivation, which may be linked to the relapsing nature of MS. In this review, the evidence for a role for viral infection in MS will be discussed with an emphasis on immune system activation related to MS disease pathogenesis.

## 1. Introduction

Multiple sclerosis (MS) is a severely debilitating progressive inflammatory disease of the central nervous system (CNS) [1]. The basic pathology is thought to be auto-immune mediated damage to the myelin sheaths of the central nerves [2]. This is supported by the finding of plaques, areas of the damage, particularly within the white matter around the lateral ventricles of the brain and optic nerves [3,4]. Demyelination of the white mater in MS is routinely demonstrated by conventional MRI techniques [5]; however, lesions in the grey matter are also demonstrated [6]. It appears that the degree of cortical demyelization reflects the clinical progression of MS with demyelination of the grey matter associated with the progressive form of the disease along with neuronal loss, while myelin destruction is detected in relapsing–remitting MS [7,8]. Cortical lesions can also be detected at the early stages and they correlate with the disease severity [9].

The clinical course of the disease varies greatly from relapsing to remitting, where patients have periods of remission, to progressive forms. There are four clinical forms of MS: primary progressive MS (PPMS), secondary progressive MS (SPMS), relapsing–remitting MS (RRMS) and progressive relapsing (PRMS), all of which are characterized by periods of active disease with evidence of new pathology interspersed with inactive periods [10] (Figure 1). RRMS is the most common form of the disease, which is characterized by worsening of clinical symptoms followed by periods of partial or complete recovery [11]. RRMS often transitions into a secondary progressive course with worsening and steady progression of symptoms [12], which is referred to as SPMS. A small group of patients will develop PPMS with steady progression of neurological symptoms without periods of remission [13,14,15]. PRMS is somewhat similar to PPMS, but these patients have periods of recovery characterized by concomitant progression of MS symptoms [15]. The remitting phase of the disease, where the periods of remission are followed by worsening of symptoms, closely resembles the progression of some viral infections, herpes viruses in particular. Although, permanent tissue destruction and loss of function is not common for reactivation of most of the herpesviruses, neurological complications have been shown in some chronic herpesvirus infections [16].

Myelin is the multilamellar sheath formed around the neurons and axons by neuroglial cells [17]. Myelin formation is a complex process requiring expression of several myelin-specific proteins: myelin basic protein (MBP), myelin-associated glycoprotein and proteolipid protein [18]. Additionally, several minor glycoproteins are present in the myelin sheath, including myelin oligodendrocyte glycoprotein (MOG) (Figure 2). MOG is expressed on the surface of the myelin, covering the neurons and axons [19]. While the function of MOG remains largely unknown, it is believed that this protein serves as an adhesion molecule or cellular receptor.

A number of risk factors, including ethnicity (particularly the HLA loci), gender (it is more common in women), latitude (and therefore sunlight and vitamin D levels) and viral infections have been identified as risk factors of MS [2]. A variety of immune modulatory treatments are used, with none fully able to halt or reverse disease progression. Nevertheless, the effectiveness of interferon beta (IFNβ) treatment of MS suggests that antiviral immunity plays a role in the etiology of MS, as this cytokine has a potent anti-viral activity [20]. A role in MS pathogenesis has been suggested for many viruses, including Epstein–Barr virus (EBV), human herpesvirus 6 (HHV-6), varicella–zoster virus (VZV), cytomegalovirus (CMV), John Cunningham virus (JCV) and human endogenous retroviruses (HERVs) [21,22,23,24,25].

The association between viral infection and MS is complex. Although belonging to different families, these viruses have in common an ability to manipulate host gene expression, potentially leading to immune dysregulation, myelin destruction and inflammation. These are all viruses with either a DNA phase or DNA viruses, which can cross the blood brain barrier (BBB) and can all establish lifelong chronic infection [26]. In this review, the role of several viruses in MS pathogenesis will be discussed.

## 2. Herpesviruses

There is an established epidemiological link between herpesvirus infection status and the risk of MS. Herpesviruses have a near ubiquitous prevalence in adult populations and are usually contracted in early childhood with little overt disease [27]. There are several herpes virus types known to be human pathogens: alpha, beta and gamma [28]. Members of each group, namely, alpha (varicella–zoster virus, VZV), beta (cytomegalovirus (CMV) and human herpesvirus 6 (HHV-6)) and gamma (Epstein–Barr virus EBV), are all suspected of having a potential role in MS. Herpesviruses can establish two replication cycles: latency and reactivation. Herpesviruses have multiple targets, including neuronal (alpha-herpesviruses), non-neuronal (beta and gamma herpesviruses), macrophages and B cells [29,30]. Herpesviruses targeting neurons directly or indirectly can contribute to tissue damage detected in MS.

Herpes viruses share many features in their structure, including the capsid and tegument proteins as well as the envelope (Figure 3). Typically, a virus genome is covered by the nucleocapsid [31], which is surrounded by the tegument protein [32,33]. The envelope containing glycoprotein spikes wraps the virus outside [34]. The envelope of glycoproteins binds to the cell receptors and assist with penetration of the target cell [35]. Virus DNA replication, transcription and encapsidation take place in the nucleus of infected cells [36,37]. In immunocompetent hosts, infection is usually asymptomatic, followed by lifelong latency and reactivation [35,38]. Viruses can reactivate, resulting in the initiation of a replication cycle and cytopathic effect in the infected cell [39].

### 2.1. Alphaherpesviruses (VZV, HSV-1 and 2)

VZV reactivation is a recognized complication of the immunosuppressive therapies used in MS treatment, in particular Fingolomid (a sphingosine-1-phosphate receptor modulator that acts by sequestering lymphocytes in lymph nodes) [40]. A history of VZV and an increased antibody response to it is more common in MS patients than the general population [40,41]. VZV is also frequently detected during the active disease phases of MS [42]. It is not clear, however, whether this detection has any connection to a role in pathogenesis in MS or is an incidental escape of VZV from immune control due to MS treatment or disease [43].

Similarly, for HSV-1 and 2, viral encephalitis as a complication of the various immunosuppressive drugs used in MS therapy is seen [44,45,46], and there has been some suggestion of increased antibody prevalence of HSV-1 and 2 in MS patients (though potentially only in some cohorts of patients [47,48]), and this is not repeatable across different cohorts of patients [49]. HSV-1 in rats and mice can induce demyelinating encephalitis but it is not clear that this cross-species transmission event pathology is relevant in humans [50,51].

### 2.2. Beta-Herpesviruses (CMV)

The association of CMV and MS pathogenesis remains inconclusive. In two studies, higher loads of CMV DNA were demonstrated in an Iranian cohort of MS patients when compared to the controls [52,53]. Corroborating these data were findings that opportunistic reactivation of CMV infection can also occur in MS patients with this reactivation potentially exacerbating existing MS [54,55]. In contrast, multiple other studies have demonstrated a negative correlation between CMV seropositivity and an MS diagnosis [24,56,57,58,59,60]. A large meta-analysis including 1341 MS and 2042 controls, however, failed to conclusively define the relationship between CMV infection and the disease [61]. These differences may potentially be explained by an effect similar to that described for Epstein–Barr virus, whereby the small number of people who have never been infected with CMV have a decreased risk of MS in contrast with reactivation of latent CMV in the active disease phase of MS, potentially exacerbating existing damage.

Evidence from the two murine models of MS is also conflicting with Pirko et al., showing a protective effect of the murine version of CMV (MCMV) infection in Theiler’s murine encephalitis virus (TMEV) model MS [62]. Whereas, Vanheusden et al. demonstrated expansion of CD4+CD28^null^ T cells in MCMV infection in mice with these cells associated with aggravation of the inflammation, demyelination and worsening symptoms of experimental autoimmune encephalomyletis (EAE), a mouse model of MS induced by the injection of myelin antigens with adjuvant EAE [63]. These authors identified circulating CD4+CD28^null^ T cells as the leading pathogenic lymphocytes in mice, as their counts correlated with demyelination and disease severity. These T cells lack the CD28 co-stimulation factor necessary for activation of T cells and are typically expanded in chronic inflammation [64]. The EAE model in mice is not, however, a perfect mirror of MS disease in humans. Although a strong correlation between CD4+CD28^null^ T lymphocytes and EAE progression has been demonstrated in mice, these cells were expanded only in a small group of MS patients and demonstrated limited autoreactivity [65]. Alternative work in the non-human primate model (the marmoset) with a closer pathology to the human disease has also highlighted that the T-cell driven responses in the murine models may not be as important in primates and humans [22].

### 2.3. Beta-Herpesviruses (HHV-6)

There are a number of studies linking HHV-6 with MS pathogenesis [66]. Strong evidence of the role of HHV-6 in MS pathogenesis includes an increased prevalence of viral DNA and proteins within MS plaques and CSF as compared to healthy patients indicating HHV-6 neurotropism [67,68]. Expression of viral RNA and proteins in periventricular lesions, which are commonly found in MS, also supports the involvement of HHV-6 in MS pathogenesis [69,70]. These findings have been countered by other studies failing to report HHV-6 detection in MS [71]. However, a recent systematic review and meta-analysis supports an association between HHV-6 antibody and DNA positivity and MS [72]. There is also some suggestion of HHV-6 proteins having cross reactivity with myelin basic protein, an essential component of the myelin sheath, which could contribute to CD8+ T cell-mediated oligodendrocyte death [71].

### 2.4. Gamma Herpesviruses (EBV)

The gamma-herpesvirus (EBV) association with MS is complex. It appears that an EBV seronegative status correlates with a decreased risk of MS [73]. Accordingly, patients with infectious mononucleosis (IM) have an increased risk of MS as compared to those who are seropositive but with no history of IM [74]. Whether the presence of EBV DNA is more likely in MS than “healthy” patients is more controversial and remains unproven [74,75,76]. Virus detection in the periphery may also not correlate with its presence in the CNS [75]. Therefore, some authors hypothesize that EBV invasion of the CNS before adaptive immune responses have developed is a crucial factor in MS pathogenesis [77]. Multiple mechanisms of EBV MS pathogenesis are currently proposed, including cross reactivity between the virus and myelin epitopes [78], auto-immune responses against alpha-β-crystallin (a stress protein expressed in lymphoid cells and oligodendrocytes) [79], antibody-dependent cell-mediated cytotoxicity and complement-dependent cytotoxicity [80]. Despite its well-established role as one of the triggers of the disease, shedding or detection of EBV in either the blood or CNS does not appear to be related to relapses or progression of MS [81,82].

Intriguingly, there is also an increasing body of evidence pointing at the role of Epstein–Barr Nuclear antigen 2 (EBNA2) in the pathogenesis of MS. EBNA2 can upregulate host gene expression and recruit transcription activation factors [83,84,85,86]. Interestingly EBNA2 binding in the host cells occurs within the known genetic loci associated with MS susceptibility [87]. In this respect, two binding sites appear to be most interesting: recombination signal binding protein for immunoglobulin kappa J region (RBPJ) and the vitamin D receptor (VDR). It has been shown that EBNA2 can convert resting B cells into immortal cells by engaging the transcription factor RBPJ [88]. These immortal B cells could maintain pathogenic autoreactive leukocytes in MS circulatory and brain tissue. The EBNA2 overlap with VDR [87] is also of importance as vitamin D deficiency as a predisposing factor in MS is well established [89]. Many of the same sites are also implicated in systemic lupus erythematosus (SLE), another disease with strong epidemiological links to EBV infection [87,90]. These associations are particularly marked in B cells and it would seem that there is a competitive interaction for transcription binding sites between EBNA2, promoting B cell proliferation and Vitamin D, which down regulates B cell function.

Further, more complicated evidence for a direct role of EBV in MS pathology is provided by the marmoset model of MS, which closely mimics the human immune response to EBV [91]. In this model, the role of Callitrichine herpesvirus 3 (CalHV3) in the pathogenesis of MS-like disease was explained by direct infection of B cells [92]. Therefore, it appears that the therapeutic efficacy of the marmoset treatment with anti CD20 monoclonal antibodies (anti B cell antibodies) was associated with the depletion of CalHV3-infected B cells [93]. An important aspect of this is the antigen presenting capacity for CalHV3-infected B cells is affected, resulting in the presentation of citrullinated epitopes of MOG, which is resistant to degradation [22]. It was suggested that these epitopes can stimulate autoreactive cytotoxic T cells, which can escape thymic deletion.

The evidence for EBV involvement in MS pathogenesis has been compelling enough for at least one trial of EBV-specific autologous T cell therapy with in-vitro expanded T cells stimulated to target EBV nuclear antigen 1 (EBNA1), latent membrane proteins 1 and 2A (LMP1, LMP2A) and reinfused in the donor patient. Seven of the 10 patients treated showed clinical and neurological improvement, though it is important to note that this was primarily a safety trial with no control arm [94].

## 3. Non-Herpes Viruses Associated with MS

### 3.1. JCV

JCV (human polyomavirus 2 or John Cunningham virus) is another near ubiquitous DNA viral infection acquired in childhood [95]. JCV is a non-enveloped double-stranded DNA virus that associates with cellular histones to form minichromosomes in infected cells [96,97] (Figure 4). It is believed that JCV infection occurs during childhood and remains dormant in the stage of latency in most individuals [98]. This explains the fact that up to 90% of adults are seropositive for the virus, with about 20% shedding it in their urine [99,100]. JCV infection does not cause overt disease in individuals with functional immune systems [101]. However, in immunocompromised individuals, the virus can trigger progressive multifocal encephalopathy (PML), characterized by lytic JCV infection of oligodendrocytes and astrocytes in the CNS [102]. It appears that the virus has to undergo several mutations to enable it to cross the BBB and replicate in the CNS [103,104].

Although JCV targets oligodendrocytes and demyelinization, it is not thought to have any role in triggering MS pathogenesis. An increased risk of development of PML in MS patients treated with natalizumab (a monoclonal antibody targeting alpha integrin and therefore inhibiting all white blood cell migration) is a known risk factor of this treatment regime [105]. Currently, the use of this drug is therefore limited to only highly active RRMS and patients with tolerance to first-line treatments such as IFN β [106]. Why this syndrome should be prevalent with natalizumab and not with other MS treatments is not clear; however, it is thought to be related to the induction of increased B cell numbers alongside reduced immune surveillance of the CNS [107]. Withdrawal of treatment can exacerbate the condition as the influx of suddenly reconstituted immune cells can worsen the inflammation caused by JCV, which is often fatal [108]. Hence, despite its effectiveness in RRMS, a risk assessment and monitoring of patients based on JCV seropositivity and antibody titer is necessary in treatment decisions with this drug in MS [23].

### 3.2. HERVs

HERVs are replication defective retroviral proviruses integrated into the human genome and comprising up to 8% of it [109]. Over the millennia, HERV proviral sequences have been integrated into the human genome regulatory machinery by functioning as promoters, repressors, poly(A) signals, enhancers and alternative splicing sites for many non-viral genes [110,111]. Along with the beneficial effects, inappropriate expression of HERVs has been shown to cause inflammation, aberrant immune reaction and dysregulated gene expression [112,113,114]. HERVs can be grouped into three main classes: Class I Gammaretrovirus- and Epsilonretrovirus-like HERVs; Class II Betaretrovirus-like HERVs; and Class III Spuma-like HERV-L [115]. Expression of Gammaretrovirus HERV family members, HERVs-H and W has been shown to be associated with an MS diagnosis [116,117]. Although not capable of completing a full replication cycle, transcription and translation of individual HERV proteins, particularly the HERV-W Env protein syncytin in the human placenta, does occur and has been demonstrated in the CNS in MS cases and in some healthy individuals [118,119,120]. There are substantial variations in the proportion of MS patients that test positive for HERV-W viral RNA in the serum, which can vary between 50 and 100% in the viral load detected [121,122,123], with our systemic meta-analysis confirming the association between MS and HERV-W expression [21]. The wide variation in HERV detection is potentially explained by population differences in HERV expression as well as the differing detection methods used in each study. It appears that the detection of HERV-W products in the blood of MS patients is associated with a poor prognosis and could serve as a predictive marker for conversion of optic neuritis into MS [124,125]. HERV load also correlates positively with Expanded Disability Status Scale (EDSS) and Multiple Sclerosis Severity Score (MSSS) ratings [126]. The higher HERV-W expression in female as compared to male patients corresponds to the gender differences within MS [125]. Further evidence of HERV association with MS pathogenesis is provided by the detection of HERV-W particles in CSF, changing with the disease progression: increasing in relapse and decreasing during remission [127]. HERV antigens can be immunogenic and higher antibody reactivity against HERV-W and HERV-H Env epitopes was demonstrated in MS patients during relapse [128]. These data suggest that HERV antigens could trigger auto-immune responses, leading to systemic activation of T cell-mediated neuropathology and brain tissue damage, as shown in a SCID mouse model [129].

There is an increasing body of data demonstrating that HERV-W protein expression leads to immune activation and inflammation. HERV-W proteins display cross reactivity with MOG and have been demonstrated to bind with the HLA DR2 locus implicated in genetic susceptibility to MS [130,131,132,133,134]. HERV-W env proteins bind to CD14 and TLR4, triggering the pro-inflammatory cytokines IL-1β, IL-6, or TNF-α [135,136,137]. The HERV-W Env-derived protein syncytin is expressed, specifically in monocytes, T and B lymphocytes and NK cells, displaying an activated phenotype with expression, increasing when these cells were stimulated with LPS. In addition, binding of syncytin activated monocytes and increased the proportion of the type of non-classical monocyte (CD14^low^CD16+) associated with MS [138]. Both HERV-W and HERV-H are overexpressed in these non-classical monocytes in MS patients [139,140]. Intriguingly the use of HERV-driven enhancers (the LTR regions in HERVs can turn on nearby genes) is increased in T cells from MS patients, specifically activating the immune genes CCL20 and IL1R2 [141]. While there is argument over whether peripheral immune responses in PBMC can induce CNS disease, it is also clear that a leaky blood brain barrier in MS can allow the migration of blood borne monocytes to the CNS, triggering inflammation and myelin damage [142].

HERV-W or syncytin (there is some argument over whether HERV-W env proteins can be reliably distinguished from each other [143]) have also been shown to inhibit oligodendrocyte precursor cell formation and remyelination, an effect that can be blocked by the anti-HERV monoclonal antibody GNbAC1 [144]. This antibody, despite a disappointing lack of effect on clinical disease scores in treatment trials with patients, did more promisingly demonstrate a reduction in new lesions as measured by MRI in treated patients compared with the placebo [145]. Recent work has in addition demonstrated that HERV-W is present in microglia (brain resident myeloid cells) associated with axons in MS patients and that expression of HERV-W in myeloid cells induces a degenerative phenotype, resulting in damage to the myelinated axons [146].

An interesting cooperation between EBV and HERVs has also been demonstrated in MS patients. Irizar et al. have shown that EBV reactivates in B cells of female RRMS patients during relapse [147]. It appears that EBV-encoded glycoprotein 350 expression stimulates the expression of the syncytin-1, HERV-W coded protein in B cells as well as in astrocytes and monocytes [147]. We have also shown a similar effect with EBV infection of B cells triggering increased expression of HERV-W RNA and protein [148]. This effect is also seen in young adults with infectious mononucleosis (EBV induced disease) [149]. It could be suggested that EBV infection or reactivation could serve as a trigger for HERV reactivation, which when acting as antigens could induce an auto-immune response targeting neural tissue.

A similar effect has been recently reported with HHV-6 infection of PBMC and astroglioblastoma cell lines where viral infection or activation of its receptor CD46 triggers HERV-W expression and TLR4 activation [150]. Similarly HSV-1 infection in neuroepithelioma cell lines with HSV-1 also activated HERV-W transcription and protein expression in neuronal and brain endothelial cells in culture [151,152], the activation potentially mediated by HSV-1 intermediate early protein (IE1) binding to the HERV-W LTR [153]. Interestingly there is also work showing that the addition of both herpes viral and HERV-H antigens to PBMC triggered enhanced cellular immune responses [154].

## 4. Antiviral Effects of MS Treatment

The treatments available for MS are all variants of immunomodulatory therapies, most of which produce their primary effect via induction of lymphopaenia or a shift to a more TH2-driven phenotype [155]. Many of them are also used in cancer therapy and common side effects include an increased incidence of opportunistic infections or reactivation of latent infections. Interestingly, the first drug successfully used in MS is IFNβ, which is also one of the principal antiviral cytokines produced by virus-infected fibroblasts [156]. It may seem a counterintuitive use of an antiviral cytokine to treat an inflammatory disorder but the feedback loops induced by IFNβ inhibit many T cell functions [155].

The more recent MS treatments include humanized monoclonal antibodies against lymphocyte surface antigens [157]. These include natalizumab that targets VLA4 (very late antigen 4), which is expressed on various leukocytes [158,159]. This is thought to inhibit the interaction between VLA-4 and vascular cell adhesion molecule-1 (VCAM-1), which facilitates leukocyte migration across the BBB [160,161,162]. However, the success of natalizumab as an MS therapy has been hindered by PML developing in some patients [108,163]. Another humanized antibody MS therapeutic is alemtuzumab, which targets CD52 expressing lymphocytes, monocytes and dendritic cells [164]. It appears that the mechanism of alemtuzumab action is associated with depletion of circulating T and B lymphocytes via antibody-dependent and complement-dependent cytolysis [164,165]. Post alemtuzumab hyper-rebounding of the B cell population can, however, result in a variety of other autoimmune diseases, a common side effect of this treatment [166]. The most recently introduced drug of this class, ocrelizumab, and its predecessor rituximab, targets the B cell surface protein CD20, resulting in selective depletion of this lymphocyte population [167]. In the context of this review, all of these therapies, which have been quite successful in MS therapy, target the immune cells in which EBV or HERV expression has been demonstrated, and part of the effect of these monoclonal antibody therapies may be in reducing the EBV and HERV-W autoreactive cells and antigen load.

## 5. Conclusions

There is increasingly solid evidence for a pathogenic role in the triggering of MS auto-immune responses by a failure to control chronic viral infections. Evidence for the herpesviruses EBV and CMV points towards patients who have never been infected with these viruses having a decreased risk of disease, whereas virus activation and the immune responses associated with them are linked to MS pathology. Similarly, EBV infection appears to trigger expression of the HERVs that have been associated with MS pathogenesis, and for both the HERVs and herpesviruses significant cross reactivity between the viral protein epitopes, MOG (myelin oligodendrocyte protein) and myelin basic proteins, which are major targets in MS autoimmunity, are evident. Directly opposing effects of vitamin D (protective) and EBV EBNA2 (associated with disease) at a molecular level are also apparent. Significantly, a number of the most commonly used and effective MS treatments also directly induce antiviral responses or remove the cells that these herpesviruses (and subsequently retroviruses) replicate and are expressed in adding further evidence to a role for these viral infections in MS pathogenesis.

## Figures and Tables

**Figure 1 viruses-12-00643-f001:**
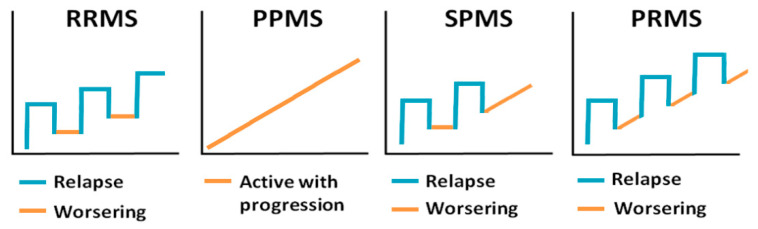
Clinical presentation of multiple sclerosis (MS). Relapsing–remitting MS (RRMS)—has worsening of clinical symptoms followed by periods of recovery; primary progressive MS (PPMS)—has steady progression of clinical symptoms; secondary progressive MS (SPMS)—initial relapsing–remitting course followed by steady progression of symptoms; and progressive relapsing MS (PRMS)—steady progression of clinical symptoms with occasional relapses.

**Figure 2 viruses-12-00643-f002:**
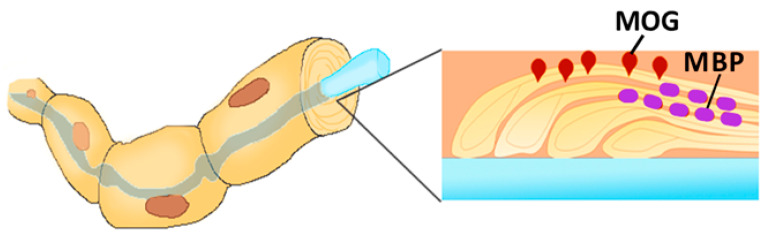
Myelin-associated glycoprotein (MOG) expression on the surface of the myelin, covering the axon. The myelin sheets are held together with Myelin basic protein (MBP), while MOG is located on the surface and exposed to the autoreactive leukocytes.

**Figure 3 viruses-12-00643-f003:**
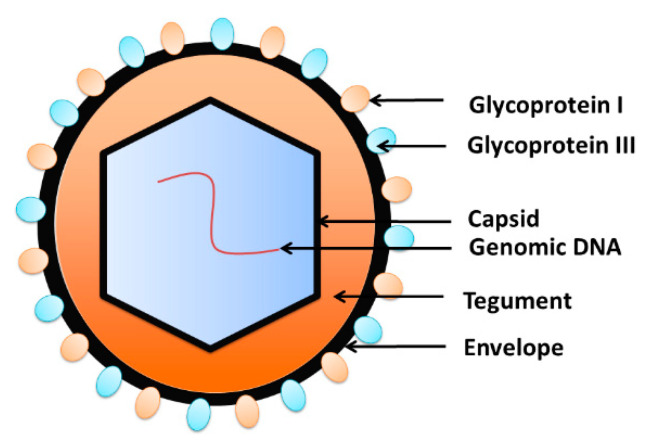
The structure of herpesviruses. The viral DNA is packed inside the capsid, which is wrapped by the tegument. The envelope, the outer layer of the virion, is composed of the phospholipids bilayer embedded with glycoproteins.

**Figure 4 viruses-12-00643-f004:**
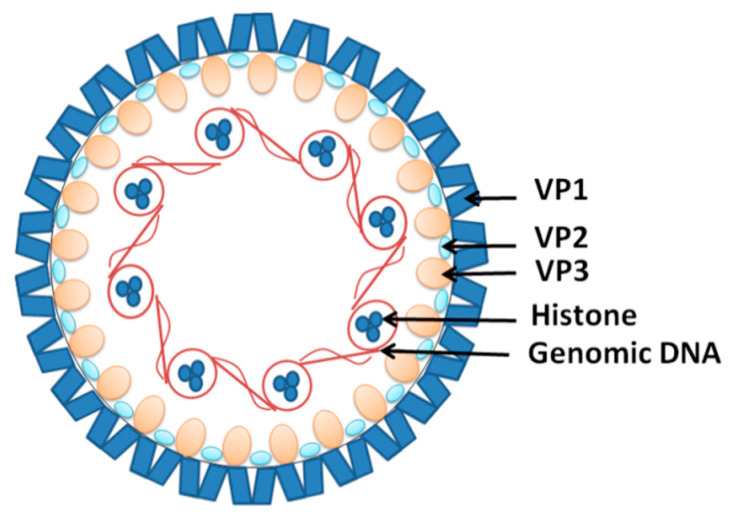
The structure of the John Cunningham virus (JCV). The viral DNA is packed around the histones in a chromatin-like complex. It is covered by viral structural proteins VP1, forming the capsid with the VP2 and VP3 proteins incorporated.

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
