# Peer review of "Role of Viruses in the Pathogenesis of Multiple Sclerosis"

_viruses, 2020, doi:10.3390/v12060643_

Round 1

Reviewer 1 Report

The submitted review article by Tarlinton et al. on viruses in multiple sclerosis (MS) is a timely summary of a subject of increasing importance and awareness. However, I currently see it as a first draft, which needs to be improved and updated (comments in order of appearance through the manuscript):

  • This manuscript should undergo a detailed spell and grammar check, also try to delete double spaces.
  • Lines 56-67 and beyond: There is much information on MOG and MOG antibodies, which to my understanding is out of the current scope of this review.
  • Reference 34 is not appropriate; in general the literature on HERVs is incomplete and needs to be updated as this field has grown substantially in the last years.
  • Line 98: Please explain “could exhaust the repair capacity of…”. This stands alone and also Fig. 3 does not refer to this proposed mode of action.
  • Line 107: Why do you introduce this subtitle here, given that under point 2 you start discussing the different herpesviruses? This structure does not make sense and must be updated.
  • Line 129: Why is HHV-6 not mentioned and discussed here?
  • Why is HSV-1 as alpha herpesvirus not presented given that there is also literature on its involvement in MS?
  • Line 158: EBV has been shown to activate HERVs and there is increasing evidence that this could contribute/mediate long-term disease development.
  • Line 195: Likewise a recent publication shows a similar interaction between HHV-6 and HERV-W (Charvet et al., 2019) as well as earlier studies already pointed to the HSV-1 and HERV activation. These must be discussed and cited.
  • Why is the chapter 4 separated and not combined in chapter 2 for beta-herpesviruses?
  • Is there any evidence on Bornavirus infection and MS (see also Feschotte et al., 2010)?
  • Lin 233: The chapter on HERVs needs to be updated and more recent references must be considered. Also please present, discuss and cite literature on these entities (particular of the type W) implicated in non-immune effects in MS (such as BBB leakage, impact on myelin repair as well as microglia activations). These are new and important insights, which must be presented.
  • Ref 138: add Mameli et al., 2013
  • Ref 144: There is more literature on TLR implication, HERV activity and glial cells, please update.
  • Lines 300-319: Too long and again out of scope of this review.

Author Response

Dear Editor,

Thank you for the chance to resubmit the manuscript. We have substantially rewritten it in response to the reviewer’s comments, with detailed responses provided below, we hope that you agree that the manuscript is much improved as a result,

Yours Sincerely,

Rachael Tarlinton

The submitted review article by Tarlinton et al. on viruses in multiple sclerosis (MS) is a timely summary of a subject of increasing importance and awareness. However, I currently see it as a first draft, which needs to be improved and updated (comments in order of appearance through the manuscript):

We would like to thank the reviewer for their comments of the timeliness of the article. We acknowledge that the presentation of the manuscript could have been better and have made the suggested changes as well as a more careful editing.

This manuscript should undergo a detailed spell and grammar check, also try to delete double spaces.

This has been done throughout

Lines 56-67 and beyond: There is much information on MOG and MOG antibodies, which to my understanding is out of the current scope of this review.

We have removed much of this as we would agree that this level of detail is not necessary for this review and some was clearly out of date as commented on by reviewer 2

Reference 34 is not appropriate; in general the literature on HERVs is incomplete and needs to be updated as this field has grown substantially in the last years.

This reference has been removed and a more appropriate range of references covering all the viruses mentioned included.

Line 98: Please explain “could exhaust the repair capacity of…”. This stands alone and also Fig. 3 does not refer to this proposed mode of action.

We have removed this section as clearly aberrant systemic immune responses resulting in autoreactive cells or antibodies crossing the blood brain barrier, with or without direct replication of the virus in the immune system are also a factor in pathogenesis and the focus of this diagram on neurological replication was not helpful. 

Line 107: Why do you introduce this subtitle here, given that under point 2 you start discussing the different herpesviruses? This structure does not make sense and must be updated.

The subtitles have been rearranged to group them more rationally into a section on herpesviruses and a section on endogenous retroviruses with the section on JCV moved to a section on viral complications of therapy (as there is not real suggestion that JCV is an initiating factor for MS)

Line 129: Why is HHV-6 not mentioned and discussed here?

The section on HHV-6 has been moved up to group with the betaherpevirus section

Why is HSV-1 as alpha herpesvirus not presented given that there is also literature on its involvement in MS?

This has been added in along with some discussion on VSV, of note the evidence for either virus being involved in MS pathogenesis is not strong but has been included for the sake of completeness

Line 158: EBV has been shown to activate HERVs and there is increasing evidence that this could contribute/mediate long-term disease development.

Yes this is discussed in the HERVs section extensively – indeed some of this is our own work.

Line 195: Likewise a recent publication shows a similar interaction between HHV-6 and HERV-W (Charvet et al., 2019) as well as earlier studies already pointed to the HSV-1 and HERV activation. These must be discussed and cited

Discussion of the Charvet et al 2018 paper has now been included and is indeed very relevant. There is also now a section summarising the work on HSV-1 and HERV activation.

Why is the chapter 4 separated and not combined in chapter 2 for beta-herpesviruses?

This has been corrected now.

Is there any evidence on Bornavirus infection and MS (see also Feschotte et al., 2010)?

No there is no evidence of bornavirus in MS. While infectious bornavirus can induce acute encephalitis in sheep, horses and humans it is confined to the range of its natural host the bicoloured white footed shrew in and near the European Alps, The plethora of reports linking bornaviruses to chronic neurological conditions like schizophrenia or MS have been shown definitively to be due to widespread laboratory contamination (Rubenstroth et al 2019 is a good review). The Feschotte report referred to by the reviewer is a review of the Horie et al 2011 report of the discovery of endogenous bornaviruses in mammalian (including human) genomes. There is no evidence linking expression of these elements to any disease at present and unlike the HERVs the endogenous bornaviruses are not thought to be able to encode proteins so are unlikely to trigger autoimmune disease.

Lin 233: The chapter on HERVs needs to be updated and more recent references must be considered. Also please present, discuss and cite literature on these entities (particular of the type W) implicated in non-immune effects in MS (such as BBB leakage, impact on myelin repair as well as microglia activations). These are new and important insights, which must be presented.

This section has been substantially updated as there has indeed been a lot of recent activity in this area

Ref 138: add Mameli et al., 2013

This has been added

Ref 144: There is more literature on TLR implication, HERV activity and glial cells, please update.

This section has also been substantially updated

Lines 300-319: Too long and again out of scope of this review.

This section has been shortened and tidied up to highlight its relevance to the main themes of the review

The section on CMV has also been extensively rewritten for clarity with much irrelevant information about murine models of MS removed.

Reviewer 2 Report

This is a useful comprehensive review of the potential role of viruses in MS pathogenesis.

Overall a balanced view is presented.

Some important work is missing: Pender M et al on EBV pathogenesis and Pilot treatments; Küry P et al on HERV, oligodendrocytes and microglia, e.g. PNAS 2019 or review in TMM.

The statements about MOG as autoantigen in MS and reference to the work by Berger et al NEJM are misleading. Subsequent work failed to replicate their findings. MOG autoantibodies are found in pediatric MS and in the separate entitiy of MOG IgG associated disease.

Overall, some more recent literature is not discussed or referenced. 

Author Response

Dear Editor,

Thank you for the chance to resubmit the manuscript. We have substantially rewritten it in response to the reviewer’s comments, with detailed responses provided below, we hope that you agree that the manuscript is much improved as a result,

Yours Sincerely,

Rachael Tarlinton

This is a useful comprehensive review of the potential role of viruses in MS pathogenesis.

Overall a balanced view is presented.

Some important work is missing: Pender M et al on EBV pathogenesis and Pilot treatments; Küry P et al on HERV, oligodendrocytes and microglia, e.g. PNAS 2019 or review in TMM.

We have included a discussion of the Pender reference of trials of EBV specific T cell therapy in the EBV section. This is indeed interesting work. The Kury reference has been added to the appropriate section on HERV induced pathology and immune responses (which has been substantially updated)

The statements about MOG as autoantigen in MS and reference to the work by Berger et al NEJM are misleading. Subsequent work failed to replicate their findings. MOG autoantibodies are found in pediatric MS and in the separate entitiy of MOG IgG associated disease.

The discussion about MOG autoantigen has been removed as we would agree that this was out of date and not necessarily

Round 2

Reviewer 1 Report

The authors have addressed most of my previous concerns and the manuscript has been improved. Last details:

Line 282: Again this is a subtitle out of range as chapter 2.1 is on alpha-herpesviruses and chapter 2.2 is on beta-herpesviruses. 

Line 481: It must reads HERVs

Line 1188: No number has been assigned to this new reference.

Line 1239: Incomplete reference.

Lines 1505-1526: No reference numbering? Where to find in the text?

Author Response

Line 282: Again this is a subtitle out of range as chapter 2.1 is on alpha-herpesviruses and chapter 2.2 is on beta-herpesviruses.

This has been corrected now.

Line 481: It must reads HERVs

This has been corrected now.

Line 1188: No number has been assigned to this new reference.

Line 1239: Incomplete reference.

This has been added

Lines 1505-1526: No reference numbering? Where to find in the text?

Now it is correct